# The Effects of Yoga Exercise on Blood Pressure and Hand Grip Strength in Chronic Stroke Patients: A Pilot Controlled Study

**DOI:** 10.3390/ijerph20021108

**Published:** 2023-01-08

**Authors:** Yen-Ting Lai, Hsiao-Ling Huang, City C. Hsieh, Chien-Hung Lin, Jung-Cheng Yang, Han-Hsing Tsou, Chih-Ching Lin, Szu-Yuan Li, Hsiang-Lin Chan, Wen-Sheng Liu

**Affiliations:** 1Department of Physical Medicine and Rehabilitation, National Taiwan University College of Medicine, Taipei 100, Taiwan; 2Department of Physical Medicine and Rehabilitation, National Taiwan University Hospital Hsin-Chu Branch, Hsinchu 300, Taiwan; 3Department of Physical Medicine and Rehabilitation, National Taiwan University Hsin-Chu Hospital, Hsinchu 300, Taiwan; 4Department of Healthcare Management, Yuanpei University of Medical Technology, No. 306, Yuanpei Street, Hsinchu 300, Taiwan; 5Department of Kinesiology, Research Center for Education and Mind Sciences, National Tsing Hua University, Hsinchu 300, Taiwan; 6Department of Pediatrics, Taipei Veterans General Hospital, Taipei 112, Taiwan; 7Faculty of Medicine, School of Medicine, National Yang Ming Chiao Tung University, Hsinchu 300, Taiwan; 8College of Science and Engineering, Fu Jen Catholic University, New Taipei City 242, Taiwan; 9Institute of Food Safety and Health Risk Assessment, National Yang Ming Chiao Tung University, Hsinchu 112, Taiwan; 10Division of Nephrology, Department of Medicine, Taipei Veterans General Hospital, Taipei 112, Taiwan; 11Department of Child Psychiatry, Chang Gung Memorial Hospital and University, Taoyuan 333, Taiwan; 12Division of Nephrology, Department of Medicine, Taipei City Hospital, Zhongxing Branch, Taipei 103, Taiwan; 13Department of Special Education, University of Taipei, Taipei 100, Taiwan

**Keywords:** stroke, yoga, hand grip strength, blood pressure

## Abstract

Background: We investigated the beneficial effect of add-on yoga with rehabilitation on blood pressure (BP) and hand grip strength in patients with chronic stroke (more than 90 days). Methods: The study included patients 30–80 years of age who could stand independently for 1 min. Patients with psychiatric diseases or undergoing other therapies (like acupuncture) were excluded. The yoga group received training (1 h session twice weekly) with standard rehabilitation for 8 weeks. The control group received standard rehabilitation only. There were no differences in age, gender, hand grip strength, or BP between the two groups (16 subjects in each group) at baseline. Results: The systolic BP (*p* = 0.01) decreased significantly, and the diastolic BP also decreased but not significantly in the yoga group (*p* = 0.11). For hand grip strength, both the unaffected hand (*p* = 0.00025) and the affected hand (*p* = 0.027) improved significantly. The control group showed no significant change in systolic or diastolic BP, nor did the grip strength change in both hands. Gender and age also affected the results of overall rehabilitation in that women benefited more from a decrease in BP, while men and young people (lower than the mean age of 60) benefited from hand grip strength improvement. Conclusions: Combining yoga with rehabilitation in chronic stroke patients can improve hand grip strength and decrease systolic BP.

## 1. Introduction

Stroke is the third leading cause of death in Taiwan, with about 11,000 deaths annually [1]. The literature indicates that hypertension is a risk factor for stroke [2,3,4]. Blood pressure (BP) control is proven to reduce the risk of stroke [5,6]. Lower blood pressure is associated with lower cardiovascular risk, especially in women [7]. Studies have also found that people with higher physical activity levels have lower rates of stroke [8]. Among adults over 45 years, lower hand grip strength is associated with a higher risk of stroke [9]. The link between grip strength and stroke risk may be due to the fact that hand grip strength is used to diagnose sarcopenia, which occurs with advancing age and has the same risk factors as stroke, such as age [9].

Patients with stroke benefit from neurorehabilitation. Hypertension is the most important risk factor for stroke. A study by Framingham shows hypertensive patients are seven times more likely to suffer from “ischemic stroke” than non-hypertensive patients [2]. Patients with a first stroke have a thirteen percent chance of recurrent stroke within one year, and this probability is 15 times compared to the general population [10]. Lowering BP is proven to reduce the risk of stroke. According to a WHO report, every 2 mmHg drop in BP can reduce the chance of stroke by six percent annually [5].

Yoga is an ancient Indian art. In Sanskrit, the word “yoga” means “union.” Yoga is a form of exercise using physical movements combined with breath regulation, postural balance, and meditation. Yoga has been extensively applied and shown positive health benefits, including BP control [11,12], improved hand grip strength [13,14], total cholesterol control [15], and adjuvant therapy for osteoarthritis and rheumatoid arthritis [16,17]. The Copenhagen Stroke Study showed that 95% of stroke patients achieved the best neurological state in the 11th week after stroke, and patients with mild stroke showed quick recovery [18]. A literature review shows that yoga is widely used as an adjunctive treatment for hypertension and hand grip strength in the general population [19,20].

To our knowledge, there is no relevant study on the relationship between blood pressure or hand grip strength in stroke patients and yoga exercise. This study aimed to investigate whether yoga combined with standard rehabilitation is more effective than standard rehabilitation alone in improving BP and hand grip strength in patients with chronic stroke. Chronic stroke is defined as a condition that occurs 3 months after an initial stroke [21].

## 2. Materials and Methods

### 2.1. Ethic Statement

The study protocol was approved by the Institutional Review Board. All patients provided written, informed consent to participate. Institutional Review Board (IRB)/Ethics Committee approval was obtained before the trial began, and the study was conducted in full compliance with the Declaration of Helsinki.

### 2.2. Inclusion and Exclusion Criteria

Eligible patients were recruited from a large teaching hospital in an urban area in the northern region of Taiwan.

The inclusion criteria were stroke ≥90 days, the ability to stand independently for 1 min [22], and aged 30–80 years.

Exclusion criteria were other complementary treatments (such as acupuncture), psychiatric disorders, use of psychotropic medications, difficulty following instructions, and other contraindications, such as cardiopulmonary disease.

### 2.3. Protocol

Between stroke onset and enrollment, eligible patients received the same rehabilitation protocol. Eligible participants were assigned to several age groups (5 years apart), and then each age group was split into two groups (first by their preference and then by assignment). The experimental group received 60 min of yoga exercise training twice a week, combined with standard stroke rehabilitation training for 8 weeks. The control group received standard rehabilitation alone (Figure 1). The yoga technique and the standard rehabilitation program were based on our previous research, described in the Appendix A [23].

### 2.4. Assessment of Outcome

The parameters were blood pressure (BP) and hand grip strength. We measured blood pressure using an electronic sphygmomanometer. Hand grip strength tests (Appendix A): the participant squeezed the dynamometer with maximum effort for at least five seconds. The grip strength was tested twice, with a minimum of 60 s rest for recovery between each attempt. Maximum grip strength was recorded as the highest value of the two tests.

One week before recruitment into the study, BP and hand grip strength were determined for both groups as a pre-test. One week after study completion, the same procedure was repeated as the post-test.

### 2.5. Statistical Analysis

Continuous data were expressed as mean ± standard deviation. The chi-square test was used for comparisons of categorical data. The *t*-test compared differences in means of continuous variables, and an independent *t*-test determined whether the means for the two groups were statistically different (indicated with dotted arrows in the figure). A paired *t*-test was used to compare the measured values before and after treatment (indicated with bold arrows in the figure).

A *p*-value of less than 0.05 was considered statistically significant (indicated with an asterisk in the figure). The sample size was based on an effective size to detect a difference between the means of the two samples. If we permitted a 5% chance of a type I error (α = 0.05) with a power of 90% and assumed the difference between the two groups was at least equal to the standard derivation, approximately 21 patients would be required. And 32 patients completed the study. For the subgroup analyses, patients were further stratified by gender and age.

The statistical analyses were performed using the SPSS for Windows statistical software package (version 19.0, SPSS, Chicago, IL, USA).

## 3. Results

There were no differences in age, gender, hand grip strength, or BP between the two groups (16 subjects in each group) at baseline (Table 1). The results showed that the systolic BP for the experimental group decreased significantly; the diastolic BP also reduced (but did not reach a significant level). However, the BP of the control group did not change significantly. Regarding hand grip strength, the yoga group showed a significant increase in the grip strength of both hands (affected and unaffected), with no significant change in the control group (Table 1) (Figure 2).

All patients (32 patients) were divided by gender (male vs. female) and age (age ≤ 60 vs. >60) sequentially to assess whether the potential impact of these factors affected the outcome of rehabilitation (Table 2).

Subgroup analysis by gender is shown in Figure 3. Women showed a significant decrease in systolic and diastolic BP after rehabilitation, which was not observed in men. However, the grip strength of the affected and unaffected hands was higher in men than in women.

Subgroup analysis by “age 60” is shown in Figure 4. As divided by the mean age (60 years), the elderly had a higher systolic BP (but not diastolic BP) than subjects below 60 years. Furthermore, younger people had higher hand grip strength in the unaffected hand throughout rehabilitation, with a significant increase in grip strength in both hands after intervention (with no significant change in both hands in people over 60).

Further linear correlations of systolic and diastolic BP and hand grip strength in the affected and unaffected hands are listed in Table 3. We found that changes in systolic and diastolic BP were highly correlated. The change in grip strength for the affected and unaffected hands also showed a high correlation (despite the effect of applying yoga or not).

Although there was a trend (*p* = 0.146) of BP change associated with hand grip strength in the yoga group, it did not reach a significant level. Thus, the change in BP was not correlated with the change in hand grip strength in this study.

## 4. Discussion

Stroke is the second leading cause of death globally and the third leading cause of disability worldwide [24]. One of the best ways to prevent stroke is to lower blood pressure [6,25]. The effect of yoga on lowering blood pressure may be related to breathing regulation and meditation techniques (mindfulness meditation, deeper/rhythmic breathing practice, and an inspiration/expiration time of around 1:2) in the yoga class [26].

The degree of disability in stroke patients is related to hand grip strength [27]. People with chronic diseases have poorer hand grip strength [28]. Yoga is proven to improve grip strength and lower blood pressure in the general population [12,13]. This finding can be attributed to the different poses, such as the cat pose and the cobra pose, extended by the tabletop pose, which requires hands and knees to support the ground, thus increasing the muscle strength of the hands through 8 weeks of add-on training [14]. Greater grip strength is associated with fewer depression symptoms, and this phenomenon is more predominant in females than males [29]. Yoga can help patients suffering with major depression [30]. Moreover, yoga interventions can reduce employees’ perceived work-related stress [31]. Our previous study showed that yoga could improve depression and balance in patients with chronic stroke [23].

Further linear regression showed the changes in systolic and diastolic BP are highly correlated, as are the changes in grip strength for the affected and unaffected hands (Table 3). However, there was no significant correlation between the change in BP and hand grip strength. Therefore, yoga may affect BP and hand grip strength through a different mechanism [26,32].

A subgroup analysis by gender (Figure 3) showed a significant drop in systolic and diastolic BP in women after rehabilitation. There are few studies on the influence of sex on BP during rehabilitation. A survey showed an increase in systolic BP of women during exercise compared to men in the past 20 years, which also needs our attention [33]. It is well established that men have stronger hand-grip strength [34]. Additionally, men had higher grip strength in the affected and unaffected hand throughout this study (Figure 3), as well as more improvement in grip strength in each hand.

As divided by mean age (60 years) (Figure 4), the elderly have higher systolic BP because of higher arterial stiffness with age [35]. People below 60 have a higher muscle body composition; therefore, the unaffected hand had higher hand grip strength throughout the study. However, the grip strengths of the affected hands were similar in both groups. Moreover, the increase in grip strength was significant after intervention in affected and unaffected hands, implying the importance of rehabilitation, especially for younger people. Hand grip strength is a significant predictor and biomarker of health [36]. Grip strength is a good prognostic factor for patients over 50 years of age with COVID-19 infection. Less admission risk associated with symptoms was noted in those with higher grip strength [37].

The literature review showed different outcomes from yoga and resistance exercise in different gender and age subgroups. A review by Inder, J.D., showed that men had a greater BP drop after isometric resistance training than women [38]. Regarding hand grip strength, work by Fritzen, A.M., compared muscle gain through resistance exercise in younger and older people in a small study (n = 18) and found no significant difference between younger and older people [39]. Therefore, the difference between yoga and resistance exercise needs further investigation to explain the difference in outcomes.

The effectiveness and safety are still in debate. As mentioned in our previous research [23], the participants had a mild disability and could stand independently for one minute. Whether yoga can lower blood pressure and improve hand grip strength in all stroke patients is still unknown. Additionally, whether yoga has an effect on the performance of other body functions (such as the Barthel Index) is also unknown. The study size was small, and the duration of beneficial effects is still unclear. The influence and benefits of yoga exercise on stroke patients warrant further large-scale studies.

A systemic review by Lawrence, M [40] shows that yoga has the potential to be a part of the rehabilitation for stroke patients. A review of research on yoga and neurological disorders by Kwok [41] showed that yoga could improve activity and quality of life and reduce anxiety and depression symptoms in patients with Parkinson’s disease. Yoga is widely used in health promotion and disease prevention and is also considered a potentially effective adjunctive treatment for neurological diseases [42]. In addition to yoga, the study has also shown that Adapted Personalized Motor Activity (AMPA) can improve the health of individuals with mental and comorbid physical disorders [43]. The AMPA system is a personalized exercise system for patients with chronic diseases [44].

## 5. Conclusions

Combining yoga with rehabilitation in patients with chronic stroke has the potential to improve hand grip strength and decrease blood pressure. Sex and age may also affect the results of rehabilitation.

## Figures and Tables

**Figure 1 ijerph-20-01108-f001:**
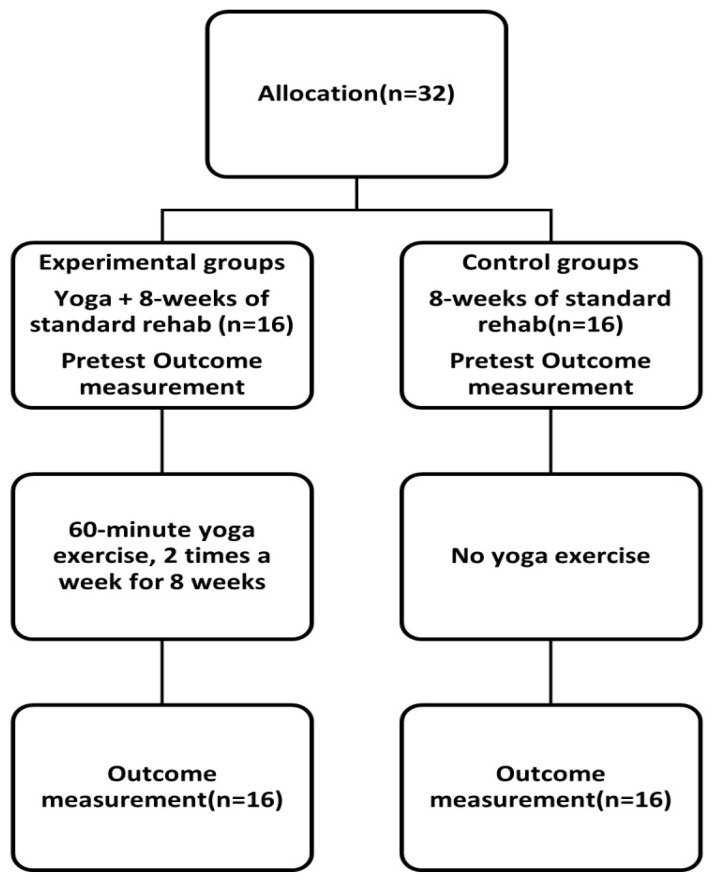
Flow diagram of the study.

**Figure 2 ijerph-20-01108-f002:**
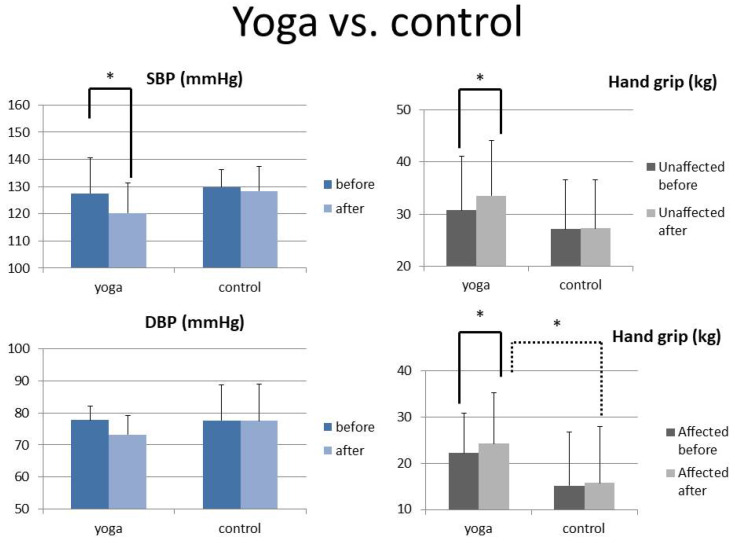
The pre- and post-intervention levels of SBP, DBP, and hand grip strength of the unaffected and affected hands between the yoga and control groups (paired *t*-test is shown with a solid line and independent *t*-test with a dotted line) *: *p* < 0.05. (Bars represent the standard deviation).

**Figure 3 ijerph-20-01108-f003:**
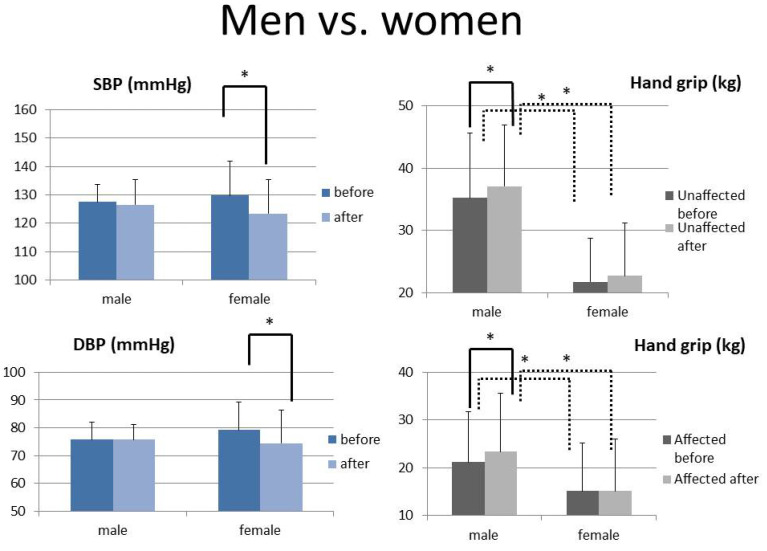
The pre- and post-intervention levels of SBP, DBP, and handgrip of the unaffected and affected hands between male and female groups (paired *t*-test is shown with a solid line and independent *t*-test with a dotted line) *: *p* < 0.05. (Bars represent the standard deviation).

**Figure 4 ijerph-20-01108-f004:**
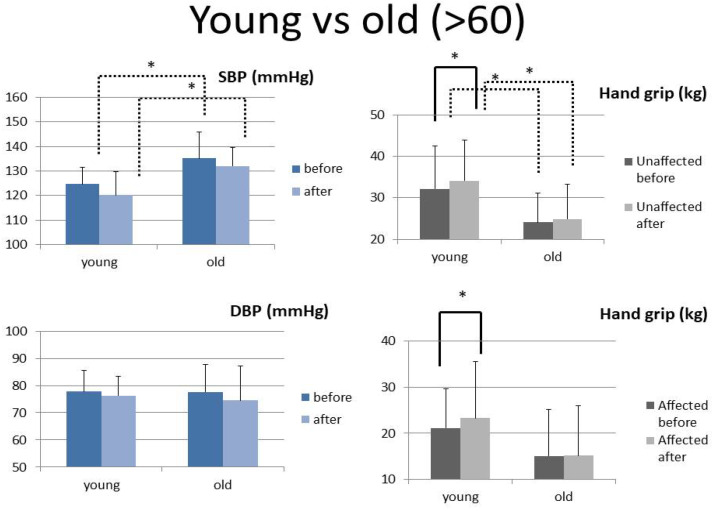
The pre- and post-intervention levels of SBP, DBP, and handgrip of the unaffected and affected hands between the young and old groups (paired *t*-test is shown with a solid line and independent *t*-test with a dotted line) *: *p* < 0.05. (Bars represent the standard deviation).

**Table 1 ijerph-20-01108-t001:** The chi-square test, blood pressure, and hand grip strength for the yoga and control groups.

	Chi-Square	Yoga+ (n = 16)	Control (n = 16)	*p*
	Sex (M:F)	10:6	8:8	0.404
	Age (≤60:>60)	11:5	8:8	0.267
Mean ± SD	All	Yoga+ (n = 16)	Control (n = 16)	*p*
**Age** (years)	59.02 ± 10.09	57.12 ± 9.12	61.19 ±10.74	0.241
T-onset (days)	382 (172–757)	360 (177.75–763.25)	418 (142–757)	0.447
T-rehab (minutes)	207.42 ± 76.09	221.25 ± 84.05	195.79 ± 68.82	0.331
**BP** (mmHg)				
Systolic BP-before	128.77 ± 9.73	127.53 ± 13.06	129.76 ± 6.61	0.642
Systolic BP-after	124.79 ± 10.50	120.27 ± 11.10	128.41 ± 8.93	0.103
**Paired t**	0.064	0.014 *	0.661	
Diastolic BP-before	77.67 ± 8.49	77.77 ± 4.42	77.58 ± 11.00	0.965
Diastolic BP-after	75.58 ± 9.44	73.18 ± 6.07	77.51 ± 11.41	0.349
**Paired t**	0.200	0.113	0.968	
**Hand grip strength** (kgs)				
Unaffected-before	28.91 ± 9.91	30.73 ± 10.34	27.11 ± 9.46	0.309
Unaffected-after	30.37 ± 10.24	33.49 ± 10.54	27.27 ± 9.23	0.086
**Paired t**	0.010 *	<0.001 *	0.837	
Affected-before	18.68 ± 10.67	22.21 ± 8.66	15.16 ± 11.59	0.061
Affected-after	20.00 ± 12.18	24.25 ± 11.03	15.76 ± 12.12	0.047 *
**Paired t**	0.012 *	0.027 *	0.249	

T-onset: time since stroke onset, T-rehab: weekly rehabilitation time. BP-before and BP-after: blood pressure before and after intervention, respectively. Unaffected-before and unaffected-after: unaffected hand grip strength before and after intervention, respectively. Affected-before and affected-after: affected hand grip strength before and after intervention, respectively. *: *p* < 0.05.

**Table 2 ijerph-20-01108-t002:** The BP and hand grip strength of subgroups before and after intervention among subgroups.

	Male (n = 18)	Female (n = 14)		Age ≤ 60 (n = 19)	Age > 60 (n = 13)	
	Mean ± SD	Mean ± SD	*p*	Mean ± SD	Mean ± SD	*p*
**BP** (mmHg)						
Systolic BP-before	127.54 ± 6.06	129.75 ± 12.15	0.647	124.66 ± 6.75	135.22 ± 10.62	0.020 *
Systolic BP-after	126.52 ± 8.73	123.41 ± 12.01	0.549	120.15 ± 9.59	132.08 ± 7.60	0.014 *
**Paired t**	0.716	0.049 *		0.104	0.409	
Diastolic BP-before	75.77 ± 6.14	79.18 ± 10.06	0.413	77.71 ± 7.75	77.60 ± 10.22	0.979
Diastolic BP-after	77.05 ± 5.44	74.42 ± 11.91	0.573	76.29 ± 7.20	74.48 ± 12.79	0.705
**Paired t**	0.539	0.042 *		0.459	0.327	
**Hand grip strength** (kgs)						
Unaffected-before	35.24 ± 9.69	21.74 ± 2.59	<0.001 *	32.19 ± 10.40	24.13 ± 7.06	0.021 *
Unaffected-after	37.13 ± 9.33	22.72 ± 3.81	<0.001 *	34.15 ± 9.77	24.85 ± 8.48	0.009 *
**Paired t**	0.020 *	0.229		0.013 *	0.364	
Affected hand-before	22.94 ± 11.37	13.86 ± 7.60	0.012 *	21.13 ± 10.65	15.10 ± 10.06	0.119
Affected hand-after	25.45 ± 12.89	13.82 ± 7.87	0.004 *	23.33 ± 12.20	15.13 ± 10.81	0.061
**Paired t**	0.003 *	0.943		0.004 *	0.959	

BP-before and BP-after: blood pressure before and after intervention, respectively. Unaffected-before and unaffected-after: unaffected hand grip strength before and after intervention, respectively. Affected-before and affected-after: affected hand grip strength before and after intervention, respectively. *: *p* < 0.05.

**Table 3 ijerph-20-01108-t003:** The linear regression of the change in blood pressure (BP) and hand grip strength.

Linear Regression R (*p*)	All	Yoga+ (n = 16)	Control (n = 16)
Delta Systolic BP vs. Delta Diastolic BP (mmHg)	R = 0.777 *p* < 0.001 *	R = 0.757 *p* = 0.011 *	R = 0.815 *p* = 0.014 *
Delta grip of the affected hand vs. Delta grip of the unaffected hand (kgs)	R = 0.587 *p* < 0.001 *	R = 0.539 *p* = 0.031 *	R = 0.638 *p* = 0.009 *
Delta BP (mmHg) vs. Delta hand grip strength (kgs)	R = 0.199 *p* = 0.429	R = 0.494 *p* = 0.146	R = 0.698 *p* = 0.054

R = coefficient of determination, * *p* < 0.005, delta systolic/diastolic BP = the change in systolic/diastolic BP after intervention, delta BP (mmHg) = delta systolic BP + delta diastolic BP, delta grip of the affected/unaffected hand = the change in grip strength of the affected/unaffected hand after intervention, and delta hand grip strength (kg) = delta grip strength of the affected hand + delta grip strength of the unaffected hand.

## Data Availability

The data presented in this study are available on request from the corresponding author. The data are not publicly available due to privacy.

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
