# Peer review of "The Effects of Yoga Exercise on Blood Pressure and Hand Grip Strength in Chronic Stroke Patients: A Pilot Controlled Study"

_ijerph, 2023, doi:10.3390/ijerph20021108_

Round 1

Reviewer 1 Report

Overall I found your study informative. Below are areas of potential improvement. 

Introduction:

Please expand on whey hand strength is tied to stroke risk.  This is an odd indicator.  What are the theories to why it is important?

Is this a preliminary study? If so, I'd include that in the introduction.

Methods:

2.2 - The wording on line 92 about "with other psychiatric disorders" is confusing.  Why were psychiatric disorder excluded.  It seems like many health conditions could impact the study, so why was this set of disorders excluded?

Figure 1 - This figure should include the 8-weeks of traditional rehab for both groups.

2.4 - Who measured the BP and the hand grip strength?  Did the patients self administer the tests?  Were these measured once before and once after 8 weeks only?  If self-administered, how do you know the tests will used correctly?

Results

Using 0s and 1s to indicate before and after treatment is confusing.  Please change the designation to B and A or just spell out before and after.  This simple change will make it much easier for the reader to understand your Tables and Figures.

Table 1 and 2 are not obvious how to interpret. Please expand your narrative in the results section to tell the reader how to interpret the table.

I like the visual representation in Figures 2, 3 and 4. Expanded narrative discussing the figures would improve the paper greatly.

Line 170: This sentence doesn't make sense.  What do you mean by subgroup analysis by "age 60"?

Discussion Line 199 and Line 205 have almost the same statement about grip strength and yoga, but cite different papers. Please remove the repeated statement.  Also, ensure that you are citing the correct references.

210: "affect" should be "affected" 

Lines 230 and 231: "Of" and "BP" are larger than the rest of the text.

Conclusions. 

With only 2 sentences, I would not recommend keeping a separate conclusion section.  Also, it seemed you had many more conclusion in the discussion / results section.  Either cut the section or add more of your conclusions.

Language

Some language issues throughout the paper do impact the ability to understand the paper. I'd recommend getting assistance with your English editing.

Author Response

Reviewer 1

Overall I found your study informative. Below are areas of potential improvement. 

 Introduction:

Q1.1 Please expand on whey hand strength is tied to stroke risk. This is an odd indicator. What are the theories to why it is important? Is this a preliminary study? If so, I'd include that in the introduction.

A1.1 Dear reviewer, lines 54-55 “Among adults over 45 years, lower grip strength is associated with a higher risk of stroke.”

This sentence is from reference 9. Liu, G.; Xue, Y.; Wang, S.; Zhang, Y.; Geng, Q. Association between hand grip strength and stroke in China: a prospective cohort study. Aging (Albany NY) 2021, 13, 8204-8213, doi:10.18632/aging.202630.

This reference speculates that the link between grip strength and stroke risk is due to hand grip strength is used to diagnose sarcopenia which occurred with advancing age, and has the same risk factor with stroke, such as age.

We have added the information to the Introduction. Thank you for your excellent suggestion.

 Methods:

Q1.2 2.2 - The wording on line 92 about "with other psychiatric disorders" is confusing.  Why were psychiatric disorder excluded.  It seems like many health conditions could impact the study, so why was this set of disorders excluded?

A1.2 Our concern was that "mental illness" might affect the impact of yoga on stroke patients. Therefore, the study excluded patients with mental illnesses to reduce interference factors. Line 94 "with other psychiatric disorders" has been corrected to "with psychiatric disorders." Thank you.

Q1.3 Figure 1 - This figure should include the 8-weeks of traditional rehab for both groups.

A1.3 Thank you for your excellent suggestion. We have revised Figure 1.

Experimental groups: Yoga +8 weeks of standard rehabilitation

Control: 8 weeks of standard rehabilitation

Q1.4 2.4 - Who measured the BP and the hand grip strength?  Did the patients self administer the tests?  Were these measured once before and once after 8 weeks only?  If self-administered, how do you know the tests will used correctly?

A1.4 Grip strength measurement: A senior physical therapist measured the patient's hand grip strength. Blood pressure measurement: Senior nurses helped patients to measure the blood pressure. These measurements were obtained once before and after 8 weeks. Thank you.

Results

Q1.5 Using 0s and 1s to indicate before and after treatment is confusing.  Please change the designation to B and A or just spell out before and after.  This simple change will make it much easier for the reader to understand your Tables and Figures.

A1.5 0s and 1s in Tables 1, 2, and Figures 2, 3, and 4 have been changed to before and after. Thank you for your excellent suggestion.

Q1.6 Table 1 and 2 are not obvious how to interpret. Please expand your narrative in the results section to tell the reader how to interpret the table.

A1.6 Tables 1 and 2 have been explained in lines 135 to 144. Thank you.

Q1.7 I like the visual representation in Figures 2, 3 and 4. Expanded narrative discussing the figures would improve the paper greatly.

A1.7 Figures 2, 3, and 4 have been explained in lines 139-141, lines 163-166, and lines 174-178. Thank you.

Q1.8 Line 170: This sentence doesn't make sense.  What do you mean by subgroup analysis by "age 60"?

A1.8 Yes. We performed subgroup analysis by "age 60." A quotation mark has been added to the age 60 of line 174 in the text. Thank you.

Q1.9 Discussion Line 199 and Line 205 have almost the same statement about grip strength and yoga, but cite different papers. Please remove the repeated statement.  Also, ensure that you are citing the correct references.

A1.9 We have deleted the repetition in line 207 and verified the references. Thank you for highlighting this error.

Q1.10 210: "affect" should be "affected" 

A1.10 Line 211 "affect" has been corrected to "affected." Thank you.

Q1.11 Lines 230 and 231: "Of" and "BP" are larger than the rest of the text.

A1.11 Thank you. We have made the necessary correction.

Conclusions. 

Q1.12 With only 2 sentences, I would not recommend keeping a separate conclusion section.  Also, it seemed you had many more conclusion in the discussion / results section.  Either cut the section or add more of your conclusions.

A1.12 Thanks for your excellent suggestion. We have incorporated the conclusions in the discussion(Lines 194).

 Language

Q1.13 Some language issues throughout the paper do impact the ability to understand the paper. I'd recommend getting assistance with your English editing.

A1.13 Thank you for the suggestion. This paper has been revised by a professional English language editing service.

Reviewer 2 Report

Line 59 - Unless in a table or graph, percent should new written in English text rather than using the symbol.

62 - Write "six percent" instead of 6%. Taking short cuts is not good English.

110 - single digits should be written in English, not Arabic.

Coordinate the Introductory statements for tables and figures should be included just before the tables and figures with a summary statement immediately following the table or figure.

214 - There was little study about the sex fac- 214 tor influence the BP of rehabilitation.  Need to rewrite this sentence.

220 - the phrase, "due to" refers to money and dates.  Betgerf to use because of or the Should that be outcomes instead of outcome?

238- one

244 - Unless there is more than one Kwok, use only last name and drop initials.

Author Response

Reviewer 2

Q2.1 Line 59 - Unless in a table or graph, percent should new written in English text rather than using the symbol.

A2.1 Dear reviewer, line 62 -13% has been corrected to thirteen percent. Thank you for your excellent suggestion.

Q2.2 62 - Write "six percent" instead of 6%. Taking short cuts is not good English.

A2.2 Line 65 -6% has been corrected to six percent. We appreciate your feedback.

Q2.3 110 - single digits should be written in English, not Arabic.

A2.3 Line 112 The single digit “5” has been rewritten to “five.” Thank you.

Q2.4 Coordinate the Introductory statements for tables and figures should be included just before the tables and figures with a summary statement immediately following the table or figure.

A2.4 Thank you for your excellent suggestion.

Q2.5 214 - There was little study about the sex fac- 214 tor influence the BP of rehabilitation.  Need to rewrite this sentence.

A2.5 Thank you for your suggestion. This paper has received professional English editing.

Q2.6 220 - the phrase, "due to" refers to money and dates.  Betgerf to use because of or the Should that be outcomes instead of outcome?

A2.6 Line 223 “due to" has been corrected to “because of." Thank you.

Q2.7 238- one

A2.7 Line 241 “1 minute” has been corrected to “One minute”.

Q2.8 244 - Unless there is more than one Kwok, use only last name and drop initials.

A2.8 Line 247 Kwok, J.Y.Y. corrected to Kwok.

Reviewer 3 Report

Dear Authors, the manuscript could provide some insight into the literature, but I find serious critical issues in the eligibility of the sample and in the selected outcomes:

-Inclusion criteria are critical.. what is meant by standing for 1 minutes, is there not even a validated score cut-off? NIH Stroke Scale? what does the exclusion criterion for other treatments mean?

-regarding the systolic blood pressure outcomes, I don't think it can give a clinical impact picture with respect to the addition or not of yoga in the rehabilitation treatment, only the hand grip partly.. Why not evaluate the Fugl-Meyer Assessment (FMA) ? Barthel Index? Trunk Control Test? 9 Hole Peg Test?

Other concerns

32 this is an aim, not a background

33 subacute

34 Inclusion criteria are critical.. what is meant by standing for 1 minutes, is there not even a validated score cut-off? NIH Stroke Scale? what does the exclusion criterion for other treatments mean?

36 what is meant by traditional?

37 is a result, put first the number of recruits with their characteristics.. regarding the systolic blood pressure outcomes, I don't think it can give a clinical impact picture with respect to the addition or not of yoga in the rehabilitation treatment, only the hand grip partly..

Why not evaluate the Fugl-Meyer Assessment (FMA) ? Barthel Index? Trunk Control Test? 9 Hole Peg Test?

89 eligibility is too non-specific, the ability to stand does not convey autonomy, among other things it could underlie massive upper limb disorders, neglect, balance disorders.. some more defined cut-off is needed.

Figure 1 is a result, you have to put only the design into the methods and then the population you wanted to reach; subsequently outcome

Age (<=60 : >60) yoga:  11:5 control:  8:8

It is absolutely not enough, this could underlie a very young population in the experimental group

Author Response

Reviewer 3

Q3.1 Dear Authors, the manuscript could provide some insight into the literature, but I find serious critical issues in the eligibility of the sample and in the selected outcomes:  -Inclusion criteria are critical.. what is meant by standing for 1 minutes, is there not even a validated score cut-off? NIH Stroke Scale?

A3.1 Dear reviewer, we were concerned about the possibility of the patient falling during exercise intervention. Therefore, the ability to stand for more than 1 minute was used as a simple inclusion criterion. Thank you.

Q3.2 what does the exclusion criterion for other treatments mean?

A3.2 It means the patient did not receive other complementary therapies to reduce the interfering factors. We have revised the text. Thank you for your suggestion.

Line 93 “Other treatments” has been corrected to “other complementary therapies.”

Q3.3 -regarding the systolic blood pressure outcomes, I don't think it can give a clinical impact picture with respect to the addition or not of yoga in the rehabilitation treatment, only the hand grip partly.. Why not evaluate the Fugl-Meyer Assessment (FMA) ? Barthel Index? Trunk Control Test? 9 Hole Peg Test? Other concerns

A3.3 Doing various tests meant arranging more funds for professional testing. Due to limited research funding, we dedicated most research grants to yoga and rehabilitation intervention courses. Thank you.

Q3.4 32 this is an aim, not a background

A3.4 Thank you for your suggestion. “Background” has been corrected to “aim.”

Q.3.5 33 subacute

A3.5 The definition of chronic stroke is based on the following study.

Cui, L.; Murikinati, S.R.; Wang, D.; Zhang, X.; Duan, W.M.; Zhao, L.R. Reestablishing neuronal networks in the aged brain by stem cell factor and granulocyte-colony stimulating factor in a mouse model of chronic stroke. PLoS One 2013, 8, e64684, doi:10.1371/journal.pone.0064684.

Q3.6 34 Inclusion criteria are critical.. what is meant by standing for 1 minutes, is there not even a validated score cut-off? NIH Stroke Scale? what does the exclusion criterion for other treatments mean?

A3.6 We were concerned about the possibility of the patient falling during exercise intervention. Therefore, the ability to stand for more than 1 minute was used as a simple inclusion criterion.

Q3.7 36 what is meant by traditional?

A3.7 Traditional refers to the standard rehabilitation program. The yoga technique and standard rehabilitation program were based on our previous research, described in the Supplement (Appendix 1 and Appendix 2). We have revised the text. Thank you for your suggestion.

Line 36 “traditional rehabilitation” has been corrected to “standard rehabilitation.”

Q3.8 37 is a result, put first the number of recruits with their characteristics.. regarding the systolic blood pressure outcomes, I don't think it can give a clinical impact picture with respect to the addition or not of yoga in the rehabilitation treatment, only the hand grip partly..

Why not evaluate the Fugl-Meyer Assessment (FMA) ? Barthel Index? Trunk Control Test? 9 Hole Peg Test?

A3.8 Thank you for your suggestion. Performing various tests meant arranging more funds for professional testing. Due to the limited availability of grants, we used most funds on yoga and rehabilitation intervention courses.

Q3.9 89 eligibility is too non-specific, the ability to stand does not convey autonomy, among other things it could underlie massive upper limb disorders, neglect, balance disorders.. some more defined cut-off is needed.

A3.9 We were concerned about the possibility of the patient falling during the yoga exercise intervention. Therefore, the ability to stand for more than 1 minute was used as a simple inclusion criterion.

Q3.10 Figure 1 is a result, you have to put only the design into the methods and then the population you wanted to reach; subsequently outcome : Age (<=60 : >60) yoga:  11:5 control:  8:8,It is absolutely not enough, this could underlie a very young population in the experimental group

A3.10 The study size was small. And “Protocol: split into two groups (first by their preference, then by assignment)”, so the age ratio of the experimental group and the control group was not consistent. This is indeed our experimental limitation. We are appreciating your comments.

Round 2

Reviewer 3 Report

Dear Authors,

I still find serious criticisms in the eligibility of the sample and in the selected outcomes:

-The inclusion criteria are fundamental.. such a wide age window is complex (30-80). I can understand the answer to my concerns about standing for 1 minute, but is there not even a validated score cut-off? Strokes can be wildly different in severity, can't you adopt an NIH Stroke Scale? In fact, the ability to stand does not convey autonomy, among other things it could underlie massive upper limb disorders, neglect, balance disorders.. some more defined cut-off is needed.

In addition, why can the blood pressure outcome give readers information about stroke rehabilitation? Instead Fugl-Meyer Assessment (FMA) ? Barthel Index? Trunk Control Test (for all asanas that stimulate the patients core)? 9 Hole Peg Test?

Table 1. Regarding the results, are you sure that a t-test is the most appropriate? there are 16 subjects, the standard deviation is large, a Wilcoxon test could provide different results. The figures with bars without standard deviation give an unclear picture, moreover, given the small sample, they should be represented with boxes and whiskers, to evaluate the interquartile range.

Does figure 4 recall the only yoga intervention? it is not clear.

Author Response

reviewer 3

Dear Authors,

I still find serious criticisms in the eligibility of the sample and in the selected outcomes:

Q1. The inclusion criteria are fundamental.. such a wide age window is complex (30-80).

A1.

Dear reviewer, setting the age width (30-80) was a concern for too few cases being accepted at first. In fact, the age of our study group was shown to be around 60 (please see the Table 1). Thank you for your excellent comments.

Q2. I can understand the answer to my concerns about standing for 1 minute, but is there not even a validated score cut-off? Strokes can be wildly different in severity, can't you adopt an NIH Stroke Scale? In fact, the ability to stand does not convey autonomy, among other things it could underlie massive upper limb disorders, neglect, balance disorders.. some more defined cut-off is needed.

A2.

  1. We are appreciating your comments. “Standing for 1 minute” we’re based on one of the evaluation items of Berg Balance Scale "STANDING UNSUPPORTED WITH FEET TOGETHER: able to place feet together independently and stand 1 minute safely."
  2. Thank you for your suggestion, and we have added the reference on line 91.

Q3. In addition, why can the blood pressure outcome give readers information about stroke rehabilitation? Instead Fugl-Meyer Assessment (FMA) ? Barthel Index? Trunk Control Test (for all asanas that stimulate the patients core)? 9 Hole Peg Test?

A3.

  1. Because blood pressure is the most important risk factor for stroke, and related studies have shown that yoga can lower blood pressure.
    Reference:
    https://pubmed.ncbi.nlm.nih.gov/33738923/
  2. Thank you for your excellent suggestion. We have complemented our research limitations on lines 243-244.
  3. The inability to perform other tests (FMA, etc.) was mainly limited by funding and time. More related research will be conducted in the future. Thank you for your valuable suggestions.

Q4. Table 1. Regarding the results, are you sure that a t-test is the most appropriate? there are 16 subjects, the standard deviation is large, a Wilcoxon test could provide different results. The figures with bars without standard deviation give an unclear picture, moreover, given the small sample, they should be represented with boxes and whiskers, to evaluate the interquartile range.

A4.

  1. About statistical distribution, the blood pressure and grip strength have been verified to be in line with the normal distribution, and the stand deviation has been added to figures 2,3, and 4 (on Lines 154, 167-168, 171-172). Because it is a before and after comparison of 32 people (16+16), the standard deviation is not large.
  2. Thank you for your excellent suggestions. We have revised our figures and made figures 2,3,4 be more clearly.

Q5. Does figure 4 recall the only yoga intervention? it is not clear.

A5.

We are appreciating your comments. All 32 people (including yoga group and no yoga group) are divided into 60 years old and above for comparison. Thank you.